# CAMK2D De Novo Missense Variant in Patient with Syndromic Neurodevelopmental Disorder: A Case Report

**DOI:** 10.3390/genes14061177

**Published:** 2023-05-28

**Authors:** Ekaterina R. Tolmacheva, Jekaterina Shubina, Taisiya O. Kochetkova, Lubov’ V. Ushakova, Ekaterina L. Bokerija, Grigory S. Vasiliev, Galina V. Mikhaylovskaya, Ekaterina E. Atapina, Nadezhda V. Zaretskaya, Gennady T. Sukhikh, Denis V. Rebrikov, Dmitriy Yu. Trofimov

**Affiliations:** Kulakov National Medical Research Center for Obstetrics, Gynecology and Perinatology, 117198 Moscow, Russia; tetisae@gmail.com (E.R.T.); jekaterina.shubina@gmail.com (J.S.); taisiya.olegovna@gmail.com (T.O.K.); l_ushakova@oparina4.ru (L.V.U.); e_bokeriya@oparina4.ru (E.L.B.); 777_gvs@mail.ru (G.S.V.); galinavalkuz@mail.ru (G.V.M.); kaatapina@gmail.com (E.E.A.); znadezda@yandex.ru (N.V.Z.); g_sukhikh@oparina4.ru (G.T.S.); d_trofimov@oparina4.ru (D.Y.T.)

**Keywords:** dilated cardiomyopathy, neurodevelopmental disorder, whole exome sequencing, WES, trio, *CAMK2D*, Ca^2+^/calmodulin-dependent protein kinase II delta

## Abstract

Background: Intellectual disability with developmental delay is the most common developmental disorder. However, this diagnosis is rarely associated with congenital cardiomyopathy. In the current report, we present the case of a patient suffering from dilated cardiomyopathy and developmental delay. Methods: Neurological pathology in a newborn was diagnosed immediately after birth, and the acquisition of psychomotor skills lagged behind by 3–4 months during the first year of life. WES analysis of the proband did not reveal a causal variant, so the search was extended to trio. Results: Trio sequencing revealed a de novo missense variant in the *CAMK2D* gene (p.Arg275His), that is, according to the OMIM database and available literature, not currently associated with any specific inborn disease. The expression of Ca^2+^/calmodulin-dependent protein kinase II delta (CaMKIIδ) protein is known to be increased in the heart tissues from patients with dilated cardiomyopathy. The functional effect of the CaMKIIδ Arg275His mutant was recently reported; however, no specific mechanism of its pathogenicity was proposed. A structural analysis and comparison of available three-dimensional structures of CaMKIIδ confirmed the probable pathogenicity of the observed missense variant. Conclusions: We suggest that the CaMKIIδ Arg275His variant is highly likely the cause of dilated cardiomyopathy and neurodevelopmental disorders.

## 1. Introduction

Dilated cardiomyopathy (DCM) is defined by the presence of left ventricular dilatation and contractile dysfunction in the absence of abnormal loading conditions and severe coronary artery disease [1]. DCM is the second most common cause of heart failure (after coronary artery disease), underlying approximately 36% of all heart failure cases, and has an estimated prevalence of over 0.4% in the general population. About 25–30% of these patients carry an identifiable pathogenic variant [2]. Many cases of DCM are familial, meaning that the variant does not occur de novo but instead cosegregates with the disease that runs in the family.

Patients with DCM are shown to receive significant clinical benefits from therapeutic approaches tailored to the underlying genetic cause [3,4]. According to different sources, nucleotide variants in over 50 genes are associated with the development of DCM [5,6] (OMIM: PS115200, accessed on 14 February 2023). None of them are associated with additional neurodevelopmental abnormalities. Most of the DCM-associated variants occur in ten genes: *TTN*, *LMNA*, *MYH7*, *MYH6*, *MYPN*, *DSP*, *RBM20*, *TNNT2*, *SCN5A*, and *TPM1* [2].

In some cases, DCM occurs as a feature of a complex phenotype comprising extracardiac symptoms, and these disorders are usually diagnosed early in life due to their severity. In such cases, a genetic cause is expected to be presented with a de novo variant (absent in healthy parents of a proband). The estimated annual incidence of DCM in children is 0.57 cases per 100,000, with an overall poor prognosis, and with 40% of children undergoing cardiac transplant or dying before 5 years post-diagnosis [7]. While the diagnostics of DCM is quite challenging because of the genetic heterogeneity and incomplete penetrance with variable expressivity described for many types of it, there are some sets of clinical features that shed light on the nature of specific DCM. For instance, in patients with Duchenne muscular dystrophy, cardiomyopathy is associated with early-onset muscle weakness, hypotonia, and calf muscle pseudohypertrophy in male patients (OMIM: 310200).

Complex syndromic forms of cardiomyopathy, and specifically DCM, are even more heterogenous and diagnostically challenging. However, the identification of the origin of the disease can be useful for the prognosis and treatment planning, as well as risk assessment for the relatives. A systematic review of the disorders classified by extracardiac features was undertaken by Lodato et al. [8]. Cardiomyopathy, dysmorphic features, and intellectual disability were selected as three major red flags for syndromic multisystemic involvement. Among those, many disorders are also associated with specific features affecting connective tissue or presenting as a progressive neuromuscular disease.

Here, we describe a child with congenital dilated cardiomyopathy and intellectual disability carrying a de novo missense variant in the *CAMK2D* gene. Considering the presence of the two above-mentioned clinical symptoms, the patient was suspected to suffer from a syndromic disorder based on the classification proposed by Lodato et al. [8]. The short life span of the patient limits the number of clinical symptoms that were observed, as other features could have manifested later in life. We aimed to investigate whether the clinical phenotype of our patient could have been caused by the *CAMK2D* variant, or whether cardiomyopathy and intellectual disability have other, possibly distinct, causes.

CaMKIIδ belongs to a family of type II multifunctional Ca^2+^/calmodulin-dependent protein kinases [9]. There are four *CAMK2* genes in the human genome, and each can yield several isoforms through alternative splicing. The four closely related genes encode isoforms: *CAMK2A*-CaMKIIα, *CAMK2B*-CaMKIIβ, *CAMK2G*-CaMKIIγ, and *CAMK2D*-CaMKIIδ. Recently, individuals carrying pathogenic variants in the genes encoding CaMKIIα and CaMKIIβ have been described [10,11]. The pathogenic variants in both genes are associated with variable neurodevelopmental disorders presenting with psychomotor delay, behavioral anomalies, variable dysmorphic features, and seizures in some patients (OMIM: 617798, OMIM: 617799). No specific cardiac anomalies were reported in these patients.

Unfortunately, the obvious limitations of our work, a single case and the presence of perinatal complications, do not allow us to draw clear conclusions, and future studies that include other patients with *CAMK2D* variants are required. However, the goal of our study was to evaluate the possibility that the above-mentioned variant is the cause of the complex disorder in our patient by exploring the existing data on the CaMKIIδ protein structure and functions and exploring the clinical phenotype of our patient in the context of phenotypes observed in the previous in vivo studies.

## 2. Materials and Methods

This case study was carried out according to the Code of Ethics of the World Medical Association (Declaration of Helsinki). The proband legal guardians (parents) gave written informed consent for the use of any data (including child photographs) for scientific purposes and publications. The study protocol was reviewed and approved by the Ethics Committee of the Kulakov National Medical Research Center for Obstetrics, Gynecology, and Perinatology (Protocol No. 7 from 17 September 2020).

The male patient came to our attention prenatally: fetal heart pathology was first detected on the 31st gestational week when a moderate dilatation of the left ventricle along with decreased contractility was described on a prenatal echocardiogram (Figure 1).

The baby was born through a C-section at 39 weeks of gestation. The parameters at birth were as follows: weight 2830 g, body length 49 cm, head circumference 35 cm, shoulder width 38 cm, and 8–9 points on the Apgar scale.

The patient became critically unstable two hours after the birth. He was on mechanical support with high-frequency oscillatory ventilation, massive inotropic therapy including Levosimendan, and inhaled nitric oxide due to pulmonary hypertension.

An ultrasound heart examination after birth confirmed the dilatation of the left ventricle, the minimum ejection fraction was 31–33%. The newborn was stabilized by the 4th day of life. Later, the child was discharged home on supportive therapy (digoxin inotropic support, therapy for pulmonary hypertension, and diuretic therapy). Pyloric stenosis was diagnosed at the age of 6 weeks of life, and surgical correction was performed. Given the sharp decrease in the left ventricle contractility that manifested prenatally, the child was diagnosed with left ventricle noncompaction. 

The patient was examined by a geneticist, with the following phenotypic features described: frontal bossing, underfolded cleft helix, prominent superior crus of antihelix, serpiginous left antihelix stem (Figure 2), single transverse palmar crease of the right hand, and dilated cardiomyopathy according to the ultrasound examination.

The complete compensation of the clinical symptoms of heart failure was achieved under drug therapy by the 7th month of life. The ejection fraction of the left ventricle was 35–37% with no sign of pulmonary hypertension. At the age of 2 years, the ejection fraction of the left ventricle reached 45%.

Neurological pathology was diagnosed right after birth with central nervous system depression and decreasing periods of wakefulness and response to examination. Neurologic status was characterized by decreased general motor activity and muscle hypotension with a delay in the formation of unconditional reflex activity. During the first year of life, the acquisition of psychomotor skills was delayed by 3–4 months, which corresponded to a moderate developmental delay. After 12 months, the formation of age-related mental functions was characterized by speech and cognitive impairments. At the age of 2 years, the child’s psychomotor development corresponded to 12–15 months and was characterized by muscle hypotension with decreased muscle strength (3 points) and symptoms of manual and general dyspraxia. The behavioral sphere was described as abnormal due to hyperkinetic disorder, signs of cognitive dysfunction, a lack of age-appropriate play activities, as well as reduced motivation for communication and cognition. His general activity was performed at a simple subject-manipulative level. At 2 years of age, the speech was represented by single simple words, simple sounds, and onomatopoeia. The overall neurological picture was dominated by problems with affective and emotional control.

Due to RSV bronchiolitis at 3 years of life, the child required hospitalization in the intensive care unit with mechanical support. It should be noted that despite the infection, there was no deterioration in cardiac function (Figure 3). After 2 months, the boy was discharged home with a recommendation of long-term inhalation therapy.

Unfortunately, 3 months after the child recovered from the RSV infection, the patient was infected with COVID-19 which became fatal. We suspect that recurrent viral infections led to the overall deterioration of the child’s condition. It is necessary to note that despite severe respiratory failure, the left ventricular ejection fraction was stable within 45% undergoing the drug therapy. However, congenital cardiomyopathy could cause significant complications of those infectious diseases. Further studies that include more patients with *CAMK2D* variants are needed for a better description of the clinical phenotype and life prognosis.

WES analysis of the proband did not reveal a causal variant, so the search was extended to trio sequencing, where a de novo missense variant (4:113513909C>T, GRCh38, c.824G>A, p.Arg275His, NM_001321571) in the *CAMK2D* gene was found. The absence of clinically significant DNA copy number variations (CNVs) was confirmed by chromosomal microarray analysis (CMA; ThermoFisher CytoScan™ Optima Suite, Thermo Fisher Scientific, Waltham, MA, USA). Whole exome sequencing was performed using a NovaSeq 6000 instrument with Illumina^®^ DNA Prep (S) Tagmentation (Illumina, San Diego, CA, USA), IDT^®^ for Illumina^®^ DNA/RNA UD Indexes, xGen Exome Research Panel v2 reagent kits (Integrated DNA Technologies, Coralville, USA; reagent kits used according to the manufacturer’s instructions). The average WES was 134.97×, 161.72×, and 128.90× (coverage at the mutation point was 41×, 52×, and 40× for proband, mother, and father, respectively). In the proband sample, the variant was identified in 17/41 reads. Bioinformatic analysis of exome data to identify causal variants was performed using proprietary software and a database. The de novo status of the variant discovered in the proband sample was established using trio analysis (confirmed by Sanger sequencing). The relationship between the proband and parents was confirmed using the plink 1.9 (https://www.cog-genomics.org/plink/) and VCFtools 0.1.16 (https://vcftools.github.io/).

## 3. Results

The analysis of the whole exome trio revealed a heterozygous Arg275His missense variant in the *CAMK2D* gene, that arose de novo (Figure 4). Pathogenicity prediction algorithms classify the variant as pathogenic (Sift: 0.02, CADD: 24.6) or benign (PolyPhen-2: 0.017). The variant was not registered in the gnomAD control database (https://gnomad.broadinstitute.org/, accessed on 14 February 2023). Moreover, according to the gnomAD, the *CAMK2D* gene is depleted in missense variants (Z = 3.11, o/e = 0.46), increasing the possibility of pathogenicity of the nucleotide variants of this type. Since, according to the OMIM database, no monogenic disorders were yet associated with the *CAMK2D* gene, we explored other known databases of genetic variants. Few missense *CAMK2D* variants were submitted to the ClinVar database (accessed on 14 February 2023) by the Laboratory of Molecular Genetics (Pr. Bezieau’s lab) (CHU de Nantes) as occurring de novo in patients with neurodevelopmental disorder; unfortunately, no additional information could be obtained. Another database gathering de novo variants in disease and control, denovo-db (https://denovo-db.gs.washington.edu/denovo-db/index.jsp, accessed on 14 February 2023), listed the Arg275His variant as occurring de novo in a patient with intellectual disability. The study that reported the variant was aimed at searching for new genes causing intellectual disability using a meta-analysis of the whole exome sequencing trios [12]. They reported ten new possible gene–disease associations; however, the *CAMK2D* gene was not among those. Taken together, database and literature search results were in favor of the pathogenicity of the variant observed in our patient.

According to the DECIPHER database (https://www.deciphergenomics.org, accessed on 14 February 2023), the region surrounding the variant has low missense constraint (between 0.2 and 0.4), further supporting the evolutionary intolerance of missense changes and high probability of their pathogenicity in this region. According to the UniProt database (https://www.uniprot.org/uniprotkb/Q13557/entry, accessed on 14 February 2023), Arg275His does not directly affect any known structural or functional domains.

CaMKII proteins play an important role in Ca^2+^ signaling throughout the body, and the CaMKIIδ isoform is predominantly found in the heart where it phosphorylates proteins involved in cardiac excitation–contraction coupling [13,14,15,16]. In the inactive state of CaMKIIδ, an inhibitory helix (αI) comprising Thr287 residue binds to the substrate-binding site preventing access to the substrates (Figure 5a). The binding of Ca^2+^/calmodulin to CaMKIIδ induces the unfolding of the αI helix, and the Thr287 residue becomes exposed to modification. The autophosphorylation of Thr287 blocks the re-binding of the αI helix, resulting in constitutive and Ca^2+^/calmodulin-independent enzyme activity [17].

To further analyze the possible deleterious effects of the Arg275 substitution, we have inspected the three-dimensional structures of CaMKII available in the Protein Data Bank. The Arg275 residue preceding the inhibitory helix is conserved across the CaMKII proteins (CaMKIIα, CaMKIIβ, CaMKIIγ, CaMKIIδ) [17]. In crystal structures of human CaMKIIδ, it is possible to adopt two different conformations. In the autoinhibited CaMKIIδ state, the Arg275 guanidinium group forms a hydrogen bond with the Gln118 side chain (Figure 5a), whereas, in the Ca^2+^/calmodulin-bound state of the enzyme, it is linked to the Trp271 main chain (Figure 5b). Interestingly, Arg275 is located at the end of the ordered region preceding the unfolded inhibitory helix in active CaMKIIδ. We consider that the structural reorganization of Arg275 facilitates the transition of the enzyme to an extended conformation: upon CaMKIIδ activation, the Arg275 side chain displaces the Ala279 residue important for the positioning of the adjacent inhibitory helix (compare Figure 5a,b).

The Arg275His variant in CaMKIIδ presumably preserves the hydrogen bond with Trp271 in the active enzyme (Figure 5b) but disrupts the interaction with Gln118 (due to the shorter side chain of His compared to Arg), thus destabilizing the autoinhibited state. It is worth mentioning that an Arg274 substitution by Glu in the close homolog CaMKIIα (corresponds to Arg275 in CaMKIIδ) resulted in an increase in Ca^2+^-independent activity [18]. Furthermore, some experimental data suggest that the Arg275His variant can increase the sensitivity of CaMKIIδ to Ca^2+^/calmodulin [19]. These data are consistent with the hypothesis that Arg275 substitutions affect the molecular regulation and activation of CaMKIIδ.

**Figure 5 genes-14-01177-f005:**
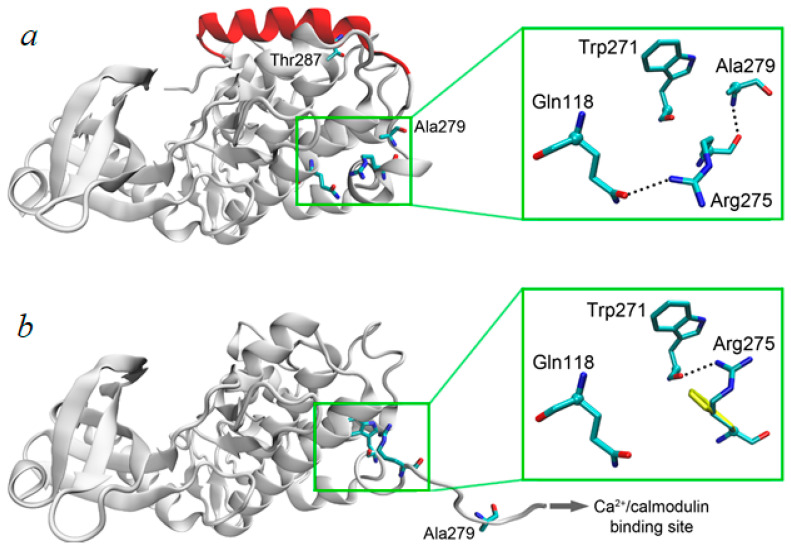
(**a**) Autoinhibited state of CaMKIIδ (PDB ID 2vn9). The inhibitory αI helix is shown in red. Ala279 forms a hydrogen bond with Arg275, which may facilitate the binding of the adjacent αI helix. (**b**) Ca^2+^/calmodulin-bound state of CaMKIIδ (PDB ID 2wel). Ala279 is displaced with the Arg275 side chain and located at an unfolded region. The Arg275His substitution, shown in yellow, was modeled with Swiss-PdbViewer [20]. Missing residues 277–280 of the unfolded region were constructed with RCD+ [21]. The figure was prepared using VMD [22].

## 4. Discussion

*CAMK2D*, the gene encoding Ca^2+^/calmodulin-dependent protein kinase II delta, is a serine-threonine protein kinase widely distributed in metazoans playing an important role in signaling within the neuronal and heart cells [9,23]. It belongs to the group of Ca^2+^/calmodulin-dependent protein kinases type II (CaMKII) that is formed by evolutionarily conserved highly homologous proteins, containing from one to four genes in eukaryotes, and four in most vertebrates [9]. Four closely related genes encode four CaMKII proteins (isoforms): CaMKIIα, CaMKIIβ, CaMKIIγ, and CaMKIIδ, the latter still not being thoroughly investigated in relation to human disease [24]. Multiple isoforms of the above-mentioned proteins acquired through alternative splicing are abundant in brain tissues and present throughout the body, playing a central role in coordinating and affecting Ca^2+^ signaling [14]. CaMKII catalyzes the phosphorylation of various classes of target proteins that explains the diversity of its effects in the body [25].

CaMKII α and β are located mostly in neurons, while the γ and δ are found throughout the rest of the body, including the heart, where isoforms γ and δ are predominant [23,26,27]. In regard to our patient, it is important to note that the CaMKIIδ protein is known to be actively expressed and enriched in both brain and heart tissues (https://www.proteinatlas.org/search/camk2d, accessed on 14 February 2023).

CaMKII oligomers can be formed from individual monomer subunits, most often in a set of 12 monomers, where catalytic domains are organized around a central connecting node [9,23]. Each CaMKII subunit consists of an N-terminal catalytic domain, an autoregulatory (autoinhibitory) domain in the middle, and a C-terminal hub domain responsible for oligomerization, as well as a linker region (also named variable domain) connecting the autoregulatory domain with the hub domain [14,28].

Normally, the autoregulatory segment is associated with the catalytic domain, keeping the enzyme catalytically inactive. When intracellular calcium levels rise, Ca^2+^ binds to calmodulin, allowing for the Ca^2+^/calmodulin complex to bind to the C-terminus of the autoregulatory segment accompanied by the unfolding of the inhibitory αI helix. As a consequence of such rearrangement, the kinase becomes activated and ready to perform its functions by the phosphorylation of other proteins in the cell [23]. CaMKIIs remain active for some period of time after the Ca^2+^ level drops, so they function as molecular switches that turn important cellular functions on or off in response to Ca^2+^ levels [17]. However, CaMKIIδ is not only regulated by the calcium level but can also affect cardiac signaling based on heart rate and action potential duration [15]. The protein activation is controlled by the frequency and duration of the Ca^2+^ spikes through the mechanism of autophosphorylation, a molecular switch that prolongs its activated state. Activated CaMKII autophosphorylates at the Thr287 of CaMKIIδ located within the αI helix. The addition of the phosphate residue disables the αI helix re-folding, creating a persistently activated or Ca^2+^/calmodulin-independent enzyme.

Regarding heart tissue, due to the fact that CaMKIIδ catalyzes the phosphorylation of various classes of target proteins, it has a unique ability to simultaneously affect both the mechanical and electrical properties of cardiac muscle cells and, accordingly, influence the development of both morphological and arrhythmic clinical phenotypes [23,25]. The hypertrophic and dilated cardiomyopathy, decreased contractility, cardiac fibrosis, pulmonary vascular congestion, and sudden death were described in mouse models with CaMKIIδ overexpression in the myocardium. The mechanism of CaMKII hyperactivity in heart failure is likely associated with the autophosphorylation mechanism—the autophosphorylated CaMKII levels were elevated in a mouse model of acute pathological left ventricular afterload and a rabbit model of left ventricular dysfunction and incessant ventricular tachycardia [25].

The expression of CaMKIIδ mRNA and protein was shown to be significantly increased in the hearts of patients with dilated cardiomyopathy compared to the hearts of donors with no structural heart defects [25]. Taken together, the evidence suggests that targeting CAMKII, specifically, the delta isoform, may provide for the selective control of heart functioning. Indeed, it was shown that the inhibition of CaMKII suppresses the development of those clinical phenotypes enhancing myocardial performance and reducing the arrhythmias rate [27,29]. Currently, CaMKII inhibition is being discussed as a promising therapeutic approach to preventing various types of cardiomyopathy [30].

In the brain, CaMKIIs account for up to 1% of the total protein in the forebrain and 2% of the total protein in the hippocampus, where they modulate processes critical to synaptic plasticity [14]. Long-term potentiation, considered to be the main cellular mechanism underlying learning and memory, is triggered by high-frequency calcium pulses leading to CaMKII activation [17]. CaMKIIs have been described as cognitive kinases due to their role in the regulation of learning and memory, as well as their autoregulatory properties, which can be considered as molecular memory [14]. It is known that CaMKII activity increases during the formation of long-term memory, and the inhibition of its activity negatively affects this process [24].

The de novo variants in the other proteins of this group (CaMKIIα and CaMKIIβ) have been clearly described in patients with neurodevelopmental disorders [10,11], while there is still some lack of evidence for the CaMKIIγ variants that were found both in patients and the control (OMIM: 618522).

The impairment of the long-term potentiation and spatial learning were described in model mice carrying a mutant protein that lacks an autophosphorylation site of the regulatory domain [14], emphasizing the importance of CaMKII autoregulation for its proper functioning.

According to the UniProt, the CaMKIIδ autoregulatory domain of the human protein is located within 283–292 amino acid residues (https://www.uniprot.org, accessed on 14 February 2023). Multiple sites in the range of 274–303 a.a. in different CaMKII subunits were found to be important for the interaction with the catalytic domain [14]. The Arg275His variant observed in our patient fell within this range.

An overview of the known CaMKII variants was recently performed to assess and describe their possible functional effects [28]. Missense changes in CaMKIIα and CaMKIIβ were shown to have either gain-of-function (activating the enzyme) or loss-of-function (reducing or silencing the kinase activity) effects depending on their location. According to our analysis of the CaMKIIδ spatial structure in its active and basal state, the Arg275His variant most probably has a gain-of-function effect making the enzyme last in the active state for a longer time and creating its Ca^2+^/calmodulin-independent form. This consideration is in line with previous studies on animal models of dilated cardiomyopathy with overexpressed CaMKIIδ. It is interesting to mention that while the z-score for missense variants is high in all CaMKII isoforms reflecting its evolutionary intolerance to such substitutions, the pLI score for the CaMKIIδ isoform equals zero, presenting its tolerance to haploinsufficiency (gnomAD, accessed on 14 February 2023). This further proves that gain-of-function is the most probable mechanism of pathogenicity.

Despite some considerable evidence for the causal role of the observed variant, the possibility that the clinical symptoms of our patient have other etiology still remains. First of all, the neonatal complications including hypoxia could have possibly caused brain damage that developed into intellectual deficiency. Second, the dilated cardiomyopathy in the fetus could also have non-genetic etiology; for instance, could be caused by some chemical or drug exposure. The other possibility is that DCM could be due to the abnormal protein function, while developmental problems of the child could be caused by perinatal brain damage that, in turn, was a consequence of heart malfunction. It should be noted that the causal role of the variant is more certain in regard to the heart phenotype being in line with the results of animal studies with CaMKIIδ overexpression [25]. However, considering the association of neurodevelopmental disorders with pathogenic variants in the two other proteins of this kinase group (CaMKIIα and CaMKIIβ), it is likely that the abnormal function of CaMKIIδ can also produce a neurological phenotype [10,11].

Functional in vitro studies could be performed in the future to confirm the role of the observed variant in the development of the clinical phenotype in our patient. The knock-in model on cardiac and neural cell culture could be helpful in evaluating the influence of the variant on cardiac cells’ mechanical and/or electrical properties as well as neuron functioning. The postmortem analysis of brain and heart tissue could also shed light on the pathophysiology of this disorder; however, in our case, it was not performed as the parents declined an autopsy. The establishment of the mechanism of pathology would serve as a basis for further studies including the exploration of treatment options. Thus, an extended study that includes more patients and in vitro experiments is required to establish the causality of *CAMK2D* variants in human pathology.

## 5. Conclusions

This clinical case demonstrates that the CaMKIIδ Arg275His mutant variant is highly likely to be the cause of dilated cardiomyopathy and neurodevelopmental disorder in our patient. While CaMKII inhibitors are actively studied as a possible treatment strategy for patients with cardiomyopathy, it might be interesting to consider whether those inhibitors could benefit patients with neurodevelopmental disorders caused by CaMKII gain-of-function pathogenic variants. Further studies describing patients with CaMKII, specifically CaMKIIδ variants, are necessary to establish its role in the monogenic form of intellectual deficiency and cardiomyopathy.

## Figures and Tables

**Figure 1 genes-14-01177-f001:**
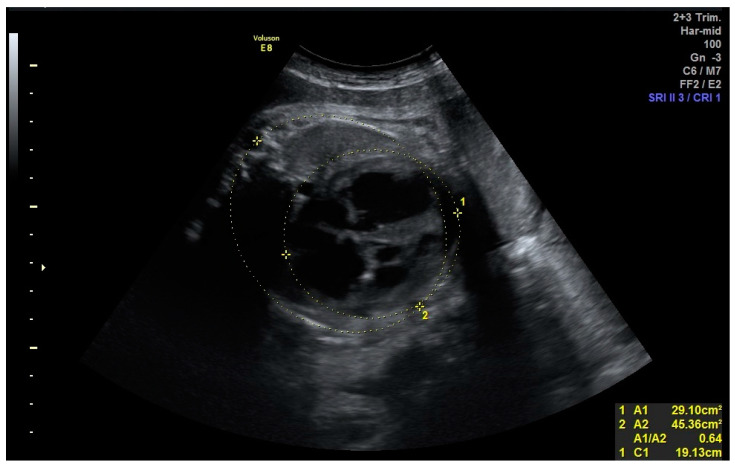
Fetal cardiothoracic ratio: 31 weeks of gestation.

**Figure 2 genes-14-01177-f002:**
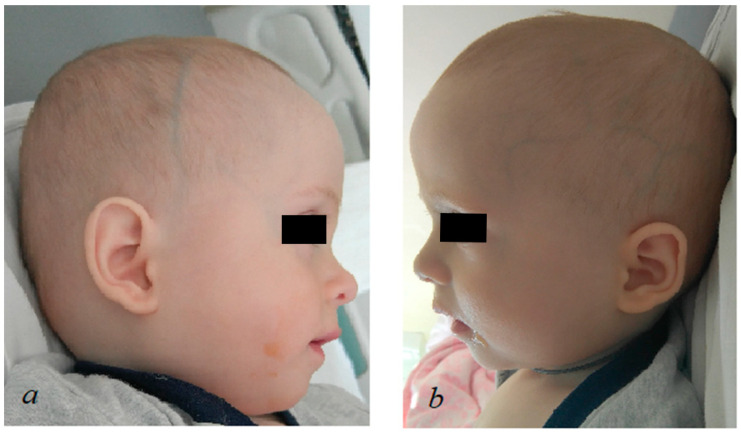
Phenotypic features at the age of 8 months of life. (**a**,**b**) Frontal bossing, underfolded cleft helix, prominent superior crus of antihelix, serpiginous left antihelix stem. The parents granted permission to publish their child’s photographs.

**Figure 3 genes-14-01177-f003:**
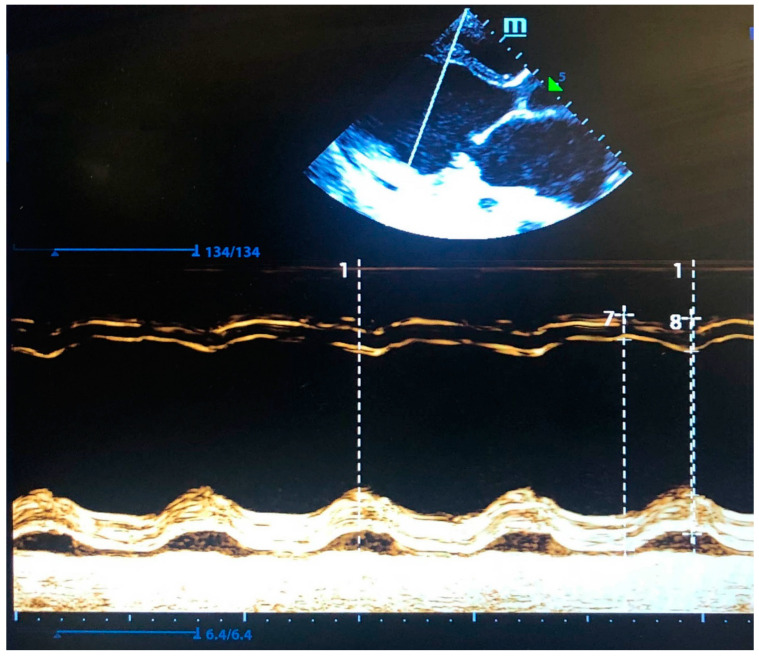
Echocardiography picture of the left ventricle at the age of 3 years (M-mode): EDVT = 139.96 mL, ESVT = 73.08 mL, SV = 66.88 mL, EF = 47.79%, FS = 24.26%, HR = 82(2) Bpm. The white numbers indicate the systolic and diastolic cardiac cycle.

**Figure 4 genes-14-01177-f004:**
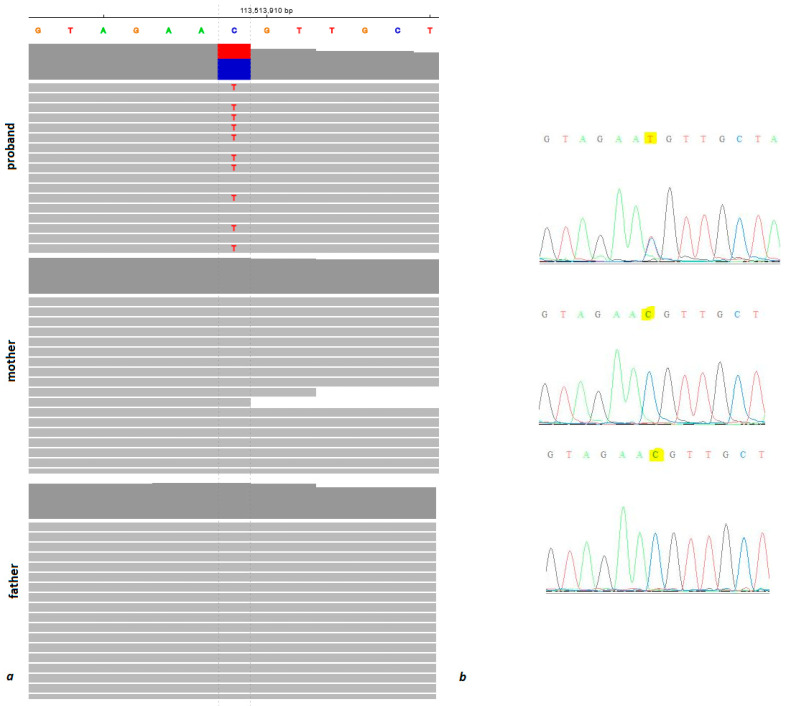
(**a**) Variant shown in IGV viewer in the proband reads and absent in the parent samples; (**b**) trio Sanger sequencing.

## Data Availability

No new data were created or analyzed in this study. Data sharing is not applicable to this article.

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
