# Peer review of "CAMK2D De Novo Missense Variant in Patient with Syndromic Neurodevelopmental Disorder: A Case Report"

_genes, 2023, doi:10.3390/genes14061177_

Round 1

Reviewer 1 Report

The authors have presented a case study of a patient displaying congenital dilated cardiomyopathy (DCM) and neurodevelopmental disorders. They did an excellent job describing the hallmarks of DCM. Additionally, they described their patient who presented with heart dilation prenatally and also suffered several complications post birth, including neurological pathology revealed through behavioral and cognitive lagging with respect to age group. The authors discovered a missense de novo mutation in the Camk2d gene, presented the lack of known evidence for this in the DCM and provided a background for its recently discovered role in a patient with intellectual disability. Furthermore, they speculate a role for this de novo mutation based on its position within the autoregulatory domain of the protein as well as cited experimental evidence. 

I have one major comment for the authors:

While the evidence presented in this case-study is compelling, the proposed mechanism of action for this mutation is speculative. For example, the authors say: "Regarding heart tissue, due to the fact that CaMKIIδ catalyzes the phosphorylation of various classes of target proteins, it has a unique ability to simultaneously affect both the mechanical and electrical properties of cardiac muscle cells, and, accordingly, influence the development of both morphological and arrhythmic clinical phenotypes" If possible, I would recommend some in vitro experiments in cardiac cells to investigate if inducing this specific mutation does indeed affect the mechanical and/ or electrical properties of cells. 

I do understand that a lot of basic science experiments that can result from this study may be out of the scope of this paper, so I am not enforcing that the above experiment be conducted. It would just be nice if the authors could at least comment on potential avenues of exploration and potential therapeutic interventions as they suggest in their manuscript. 

The manuscript was clearly written and easy to understand. There were only a few grammatical errors that I noticed such as 

1. An unnecessary comma in the first line of the introduction: "Dilated cardiomyopathy (DCM) is defined by the presence of left ventricular dilatation and contractile dysfunction, in the absence of abnormal loading conditions and severe coronary artery disease"

2. Awkward phrasing in the abstract: "Trio sequencing revealed a de novo missense in the CAMK2D gene (p.Arg275His), while this gene is not currently associated with any specific inborn disease"

Author Response

We are grateful to the reviewers for the time devoted to dive into our study. Here are the answers to the questions and comments that should make the work more clear. Thank you!

Reviewer 1

Q1) It is advisable that the authors include the study design type within the title.

A1: The study design type is added in the title

Q2) The authors should work with English editing services to improve readability of the text.

A2: English language corrected where cockney did not match

Q3) I advise the authors to more clearly state the research question and their goals in the last introduction paragraph.

A3: The goal of the study was to investigate whether the observed de novo variant in a gene that was not previously described as associated with monogenic disorders could be the cause of the clinical symptoms of our patient. We have added the lines stating that, and also the obvious limitations of our study.

Q4) I advise the authors to revise their introduction, especially the 5th paragraph. For example, the authors state “Despite a high probability of having a syndromic disorder based on the above-mentioned classification, our patient was not suspicious for any known specific type of multisystemic DCM-based disease”. However, the lifespan of the child was considerably short ad perhaps if the child had lived longer, signs of multisystemic disease would have appeared. I think this should be considered in the rationale of the authors.

A4: Thank you for the comment. We definitely agree that the child’s short life span limits the number of possible clinical symptoms that could have manifested later in life. It was our intention to state that the symptoms that we observed did not allow for a clear diagnosis but we suspected it could be syndromic because of the simultaneous presence of both DCM and intellectual disability. We have revisited the paragraph to make it more clear.

Q5) Also, I advise the authors to restructure this sentence “We believe that the coexistence of neurological symptoms with DCM in our patient is not a coincidence”. Following this sentence an explanation or hypothesis is required.

A5: Yes, this statement requires more explanation so we have excluded it from the introduction to revisit later in Discussion.

Q6) Considering the critical events that occurred right after the birth of the subject, couldn’t the hypoxia, mechanical ventilation, etc. have caused the neurodevelopmental abnormalities in this case?

A6: We included that in the discussion.

Q7) Were any postmortem analysis performed?

A7: Unfortunately, not. The parents declined autopsy.

Q8) In my view, considering that not always a correlation means causality, it would be essential in this study a neuropathological evaluation postmortem to understand the cause of developmental disorder in this case. It could be related to a gene, but it is impossible to ignore the events that occurred after birth as a cause of developmental disorder.

A8: That is true. We have included this consideration in the Discussion section.

Q9) The discussion section should include limitations of the study and explore additional causes for developmental disorder in this particular case, such as hypoxia related to the DCM, hypoxia in birth,etc.

A9: That is true. We have included this consideration in the Discussion section.

Reviewer 2 Report

Editing required.

Author Response

We are grateful to the reviewers for the time devoted to dive into our study. Here are the answers to the questions and comments that should make the work more clear. Thank you!

Reviewer 2:

The authors have presented a case study of a patient displaying congenital dilated cardiomyopathy (DCM) and neurodevelopmental disorders. They did an excellent job describing the hallmarks of DCM. Additionally, they described their patient who presented with heart dilation prenatally and also suffered several complications post birth, including neurological pathology revealed through behavioral and cognitive lagging with respect to age group. The authors discovered a missense de novo mutation in the Camk2d gene, presented the lack of known evidence for this in the DCM and provided a background for its recently discovered role in a patient with intellectual disability. Furthermore, they speculate a role for this de novo mutation based on its position within the autoregulatory domain of the protein as well as cited experimental evidence.

Q1) I have one major comment for the authors:

While the evidence presented in this case-study is compelling, the proposed mechanism of action for this mutation is speculative. For example, the authors say: "Regarding heart tissue, due to the fact that CaMKIIδ catalyzes the phosphorylation of various classes of target proteins, it has a unique ability to simultaneously affect both the mechanical and electrical properties of cardiac muscle cells, and, accordingly, influence the development of both morphological and arrhythmic clinical phenotypes" If possible, I would recommend some in vitro experiments in cardiac cells to investigate if inducing this specific mutation does indeed affect the mechanical and/ or electrical properties of cells.

I do understand that a lot of basic science experiments that can result from this study may be out of the scope of this paper, so I am not enforcing that the above experiment be conducted. It would just be nice if the authors could at least comment on potential avenues of exploration and potential therapeutic interventions as they suggest in their manuscript.

A1: Thank you very much for your comment and valuable ideas. It is true that functional studies in both cardiac and neural cells would significantly strengthen the results of our work. Moreover, we aimed to perform those in collaboration with another scientific group as we believe the study would benefit from the comparison of different patients with CAMK2D variants. Hopefully, this can be done in the future within the frame of an expanded investigation that would include more patients.

Round 2

Reviewer 2 Report

The authors have replied to all of the reviewer's comments sufficiently.

Minor English editing required.